# Adversarial Robustness via Deformable Convolution with Stochasticity

**Yanxiang Ma** [* 1]  **Zixuan Huang** [* 2]  **Minjing Dong** [3]  **Shan You** [4]  **Chang Xu** [1]

## Abstract

Random defense represents a promising strategy to protect neural networks from adversarial attacks. Most of these methods enhance robustness by injecting randomness into the data, increasing uncertainty for attackers. However, this randomness could reduce the generalization capacity of defense, as defense performance could be sensitive to the hyperparameters of noise added to the data, making it difficult to generalize across different datasets. Additionally, the involvement of randomness always comes with a reduction of natural accuracy, which leads to a delicate trade-off between them, which is seldom studied in random defense. In this work, we propose incorporating randomness into the network structure instead of data input by designing stochastic deformable convolution, where a random mask replaces the convolutional offset. This process promotes data independence, enhancing generalization across datasets. To study the trade-off, we conduct a theoretical analysis of both robust and clean accuracy, from a perspective of gradient cosine similarity and natural inference. Based on the analysis, we reformulate the adversarial training in our random defense framework. Extensive experiments show that our method achieves SOTA adversarial robustness and clean accuracy compared with other random defense methods. Code is available here.

## 1. Introduction

Despite the remarkable success of deep neural networks (DNNs) in classification tasks (Park et al., 2022; He et al., 2016; Zagoruyko & Komodakis, 2016; Dosovitskiy et al.,

2020), they remain susceptible to subtle adversarial perturbations, which can mislead them into making incorrect predictions (Madry et al., 2017; Szegedy et al., 2013a; Dong et al., 2017; Moosavi-Dezfooli et al., 2016; Croce & Hein, 2020). There have been various approaches explored to improve model robustness (Zhang et al., 2019; Wang et al., 2020; Chakraborty et al., 2021; Xie et al., 2017; Carbone et al., 2021; Li et al., 2019; Cohen et al., 2019; Dong et al., 2022; Fu et al., 2021; Ma et al., 2023; Panousis et al., 2021). Among them, random defense methods (Xie et al., 2017; Carbone et al., 2021; Li et al., 2019; Cohen et al., 2019; Dong et al., 2022; Fu et al., 2021; Ma et al., 2023; Panousis et al., 2021) shows notable effectiveness due to their randomness that disrupts adversarial strategies.

Existing random defense algorithms widely adopt additive noises to involve randomness, which makes the defense performance sensitive to the hyperparameters of noises. For example, some random defense methods add noises to the data or feature maps in the networks to achieve adversarial robustness (Xie et al., 2017; Carbone et al., 2021; Ma et al., 2023; Li et al., 2019). However, the noise distribution, variance, *etc*., always requires careful tuning to achieve satisfactory performance, which makes it difficult to generalize to different datasets. Meanwhile, some random defense methods (Dong et al., 2022; Xie et al., 2017; Carbone et al., 2021; Panousis et al., 2021) focus on reducing adversarial transferability among different sampled paths, there is no in-depth analysis of the trade-off between natural accuracy and robustness, even though the clean accuracy could be significantly influenced by the introduced randomness.

In this paper, we incorporate randomness into the network structure instead of data input to achieve a framework with data independence and generalization across datasets. Considering the commonality of convolutional layers in neural networks, we cooperate with deformable convolutions (Dai et al., 2017; Zhu et al., 2018) and replace the fixed offsets with random masks to design a **D**eformable **C**onvolution with **S**tochasticity (DCS), which effectively reduces the similarity of the gradient in a data-independent way. Based on our designed framework, we theoretically analyze the trade-off between robustness and clean accuracy from the perspectives of gradient cosine similarity and natural inference. On the one hand, we derive an upper bound for the kernel size to satisfy gradient cosine similarity constraints.

---

[*]Equal contribution [1]School of Computer Science, University of Sydney, NSW, Austrilia [2]International School, Beijing University of Posts and Telecommunications, Beijing, China [3]School of Computer Science, City University of Hong Kong, Hong Kong, China [4]SenseTime Research, Beijing, China. Correspondence to: Chang Xu <c.xu@sydney.edu.au>.

*Proceedings of the 42$^{nd}$ International Conference on Machine Learning*, Vancouver, Canada. PMLR 267, 2025. Copyright 2025 by the author(s).

On the other hand, we calculate a lower bound for the kernel size using the distance between two predictions to enhance clean accuracy affected by the introduced randomness. By demonstrating the data independence of both bounds, we selected an optimal kernel size at their intersection, establishing our data-independent framework. Finally, by adaptively optimizing the DCS during AT to remove points with similar gradient sources, we further enhance the robustness of our proposed framework. Our work has the following main contributions:

**1.** We design a data-independent framework called DCS to achieve random defense against adversarial attacks.

**2.** We theoretically explore how kernel size manages the trade-off between natural accuracy and gradient similarity, keeping it within a data-independent range.

**3.** We design a gradient-selective adversarial training (GSAT) algorithm to remove points with similar gradient origins in DCS, reducing attack transferability.

## 2. Related Works

Deep Neural Networks (DNNs) are vulnerable to adversarial perturbations (Goodfellow et al., 2014a; Szegedy et al., 2013b), which results in the development of attack and defense algorithms (Szegedy et al., 2013a; Madry et al., 2017; Croce & Hein, 2020; Carbone et al., 2021; Li & Xu, 2023; Dong et al., 2020; Cheng et al., 2023; Dong et al., 2021; 2022; Li et al., 2019; Panousis et al., 2021; Mei et al., 2025; Gong et al., 2024a;b; Bartoldson et al., 2024; Chen & Lee, 2024; Amini et al., 2024; Moosavi-Dezfooli et al., 2016; Wang et al., 2023; Carlini & Wagner, 2017; Dong et al., 2017). Commonly, the white-box attacks generate adversarial examples via gradients on the input (Szegedy et al., 2013a; Dong et al., 2017; Madry et al., 2017; Croce & Hein, 2020; Athalye et al., 2018b). These algorithms design very small adversarial perturbations along the direction of gradient ascent. The white-box attacks are very hard to detect but do a lot of harm to network performance. In recent years, to defend against white-box attacks, several randomized-based algorithms have been proposed to defend against adversarial perturbations (Chakraborty et al., 2021; Carbone et al., 2021; Dong et al., 2022; Dong & Xu, 2023; Li et al., 2019; Panousis et al., 2021; Ma et al., 2023; Mei et al., 2025; Gong et al., 2024a;b; Bartoldson et al., 2024; Chen & Lee, 2024; Amini et al., 2024). Some of the works inserted extra random layers with random weights and multiplied them with a specific feature map during the forward propagation (Chakraborty et al., 2021; Carbone et al., 2021). (Ma et al., 2023) designed a trainable randomized layer that fits the network through training, to optimize the effect of randomness. Additionally, the random parameters can also be added to the feature map as a regularization term (Li et al.,

2019). Other works added randomness in the network structure by randomly choosing different normalization methods (Dong et al., 2022). Later, (Panousis et al., 2021) adds randomness to the parallel random convolutional layer and then filters the multiple outputs to compose a new feature map. This method introduces extra parameters which are expensive in training. Furthermore, these structural randomization methods are not pure structural randomization. The weights of each randomized structure are independent, which is effectively mixed structure-weight randomization. Adversarial Training (AT) (Madry et al., 2017) is a training method inspired by adversarial generative networks, which is the most commonly used training method for adversarial robust models. To prevent overfitting in this process, (Rice et al., 2020) proposes an early-stop algorithm. AT is widely used in many adversarial defense methods (Chakraborty et al., 2021; Carbone et al., 2021; Dong et al., 2022; Dong & Xu, 2023; Li et al., 2019; Panousis et al., 2021; Ma et al., 2023; Mei et al., 2025; Gong et al., 2024b).

## 3. Preliminary

### 3.1. Deformable Convolution

To tackle complex spatial variations in data, Deformable Convolution Networks (DCNs) are proposed to replace the normal convolution layer with a deformable convolution layer (Dai et al., 2017; Zhu et al., 2018). In the deformable convolution layer, an offset varies the location of points in the kernel to deform it. The offset can be considered as a binary mask, which masks out the original location and exposes a new location. With this thought, DCN first defines an initial convolutional kernel $\mathcal{K}$ whose kernel size is a learnable variable $k_0$. Then we determined a collection of masks $\mathbb{M}^{k \times k}$ where every mask $M \in \mathbb{M}$ passes $n$ points. The deformed kernel $\mathbf{K}$ can then be built by $\mathbf{K} = M \odot \mathcal{K}$. The dimension of deformed kernel $\mathbf{K}$ is defined as $n \times C_{in} \times C_{out}$, where $n$ is the receptive field of the deformed kernel. In this paper for variable convolution, kernel K is considered as a collection of points. By these definitions, DCN can be pixel-wised defined using masks as

$$y_l = DCN(l) = \sum_{K_i \in \mathbf{K}} w(K_i) \cdot X_{i+l}, \qquad (1)$$

where $l$ is the position of kernel performed on feature map $X$. In DCN, $k_0$ is a learnable variable, while in our method $k_0$ is fixed onto a certain value $k$.

### 3.2. Randomized Adversarial Defenses

We denote $h \in \mathbb{H}$ as the classifier that maps the input $\mathbf{X} \in \mathbb{X}^{H \times W \times C}$ to the logits $h(\mathbf{X})$, where $\mathbb{H}$ is the hypothesis space of image classification models and $H \times W \times C$ is the size of the input image. In white-box settings, adversarial attacks maximize the loss function $\mathcal{L}(h)$ by adding an

adversarial perturbation $\Delta$ to $\mathbf{X}$, which forms adversarial examples (AEs) $\tilde{\mathbf{X}}$. Given the maximum perturbation size $\epsilon_d$ and the true label $\mathbf{y}$, AE can be defined as,

$$\tilde{\mathbf{X}} = \underset{\tilde{\mathbf{X}}:\|\tilde{\mathbf{X}}-\mathbf{X}\|_p \leq \epsilon_d}{\arg\max} \mathcal{L}(h(\tilde{\mathbf{X}}), \mathbf{y}), \quad (2)$$

where $\Delta$ is constrained by $\epsilon_d$ via $l_p$-norm. While adversarial attacks maximize the loss function, adversarial training (AT) minimizes it. By using the predefined adversarial example as training data, AT finds $h^*$ that satisfies,

$$h^* = \underset{h \in \mathbb{H}}{\arg\min} \mathbb{E}_{\mathbf{X},\mathbf{y} \sim \mathbb{X},\mathbb{Y}}[\mathcal{L}(h(\tilde{\mathbf{X}}), \mathbf{y})]. \quad (3)$$

Different from adversarial training, randomized defenses prevent attackers from generating accurate AEs (Chakraborty et al., 2021; Carbone et al., 2021; Dong et al., 2022; Fu et al., 2021; Ma et al., 2023). Given two classifiers $h_1$ and $h_2$ randomly sampled from subset $\mathcal{H} \subset \mathbb{H}$, the defense scheme is formulated as

$$\mathcal{H}^* = \underset{h_2 \in \mathcal{H}}{\arg\min} \mathbb{E}_{\mathbf{X},\mathbf{y} \sim \mathbb{X},\mathbb{Y}}[\mathcal{L}(h_2(\tilde{\mathbf{X}}'_1), \mathbf{y}], \quad (4)$$

where $\tilde{\mathbf{X}}'_1 = \arg\max_{\tilde{\mathbf{X}}'_1:\|\tilde{\mathbf{X}}'_1-\mathbf{X}\|_p \leq \epsilon_d} \mathcal{L}(h_1(\tilde{\mathbf{X}}'_1), y)$. In $h_1$ and $h_2$, The parameters or structures $h_2$ vary from those in $h_1$, which lead to different forward paths. White-box attack settings can be thus downgraded to black-box settings since $h_1$ and $h_2$ are randomly sampled. The objective is thus converted to the reduction of adversarial transferability between $h_1$ and $h_2$, which achieves adversarial defense.

## 4. Methodology

### 4.1. Deformable Convolution with Stochasticity

Randomized defense methods are always sensitive to their hyperparameters about randomness. In existing works, most of the bounds on hyperparameters are data-dependent (Ma et al., 2023; Dong et al., 2022; Li et al., 2019; Carbone et al., 2021). When applying these defense methods to unknown data, the hyperparameters need to be fine-tuned via a large amount of observation experiments. To address this limitation, we incorporate randomness into the network structure, isolating the hyperparameters that control randomness from the data flow in the pipeline. To make structural randomness generalizable across different CNNs and their variants, we randomize convolution layers via deformable convolution.

Specifically, we propose Deformable Convolution with Stochasticity (DCS), which accomplishes structural randomized defense by constructing a random space for deformed convolution kernels. In detail, following the definition of Sec. 3.1, we randomly sample some masks from $\mathbb{M}$ to construct a random space for the deformed kernels $\mathbb{K}^{n \times C_{in} \times C_{out}}$, where all the kernels in the random space

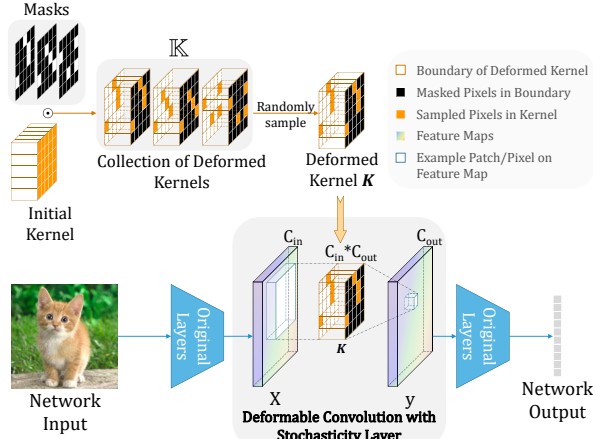

Figure 1. This figure shows how DCS performs random defense on a predefined network. The top half of this figure shows the process by which DCS generates a collection of deformed kernels and the internal situation of the deformed kernel. The lower half shows the pipeline of a network containing a DCS, where the DCS layer is shown specifically.

distribute uniformly. In each of the network forward propagation, DCS samples a deformed kernel $\mathbf{K} \sim \mathbb{K}$. The structure of DCS in a predefined network is shown in Fig. 1

In DCS, the hyperparameters that control randomness are the receptive field of the deformed kernel $n$, the initial kernel size $k$, and the stride $S$. To keep the dimension of the output feature maps consistent with the demand of the network data flow, $S$ keeps consistent with the original convolutional layers. $k$ is close to the kernel size of the original convolution layer, varying padding to fit the demand of output dimension. Thus, randomness can only be controlled via $n$, which is independent of data. Thus, DCS solves the limitation on the randomness of the data sensitivity. Based on the description above, we define the function of DCS as

**Definition 1.** *Denote $\mathbf{K}_l \sim \mathbb{K}_l$ a sampled deformed kernel at location $l$, whose input feature map is $X$ and output feature map is $y$. Denote $X_i$ the pixel on $X$ at location $i$, and $K_i$ the point on $\mathbf{K}_l$ at location $i$. Following the definition of DCNs in Eq. 1 we define DCS on the pixel level as,*

$$y_l = DCS(l) = \sum_{K_i \in \mathbf{K}_l} w(K_i) \cdot X_{i+l}, \quad (5)$$

*where the receptive field of $\mathbf{K}$ at each channel is $n$.*

Given definition 1, we can regard normal $m*m$ convolution as a special case for DCS, where $k = n = m^2$. Replacing a normal convolution with DCS can be regarded as generalizing a special case into a normal one. Thus, DCS can replace any convolutional layers theoretically, which provides a high DCS generality of DCS among convolution-based networks.

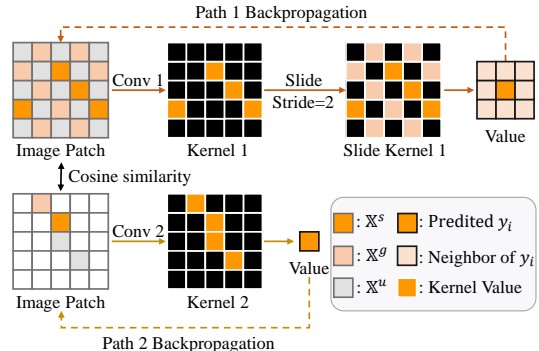

*Figure 2.* An illustration of three different cases corresponding to the three terms in Eq. 7. The upper Path 1 represents the attacked path and the lower Path 2 shows the inference path, where the dark orange, light orange, and gray points represent points in the collection $\mathbb{X}^s$, $\mathbb{X}^g$ and $\mathbb{X}^u$, respectively.

### 4.2. Transferability Analysis of Random Kernels

The defined DCS introduces different inference paths. To increase its robustness, we reduce the adversarial transferability between different inference paths by controlling their gradient similarity on DCS input feature maps $X$, using the receptive field $n$. The gradient similarity on the input feature map between each inference path can be defined as

$$\text{Cos} < \nabla_X, \nabla_{X'} > = \frac{\nabla_X \cdot \nabla'_X}{\|\nabla_X\|\|\nabla'_X\|}, \qquad (6)$$

where $\nabla_X$ is the gradient of the example that the attacker can access, and $\nabla'_X$ is the gradient that the network will have in inference. A small gradient similarity represents that it is hard for the attacks to transfer from the attacked path to the inference path. For a detailed analysis of $n$, we downgrade the gradient similarity to pixel-level. Specifically, we first divide the points on the input feature map into three types. In the first inference, we define $\mathbb{X}^s$ the collection points selected by the kernel from location $i$, $\mathbb{X}^g$ the collection points not in $\mathbb{X}^s$ but are selected by kernels from locations other than $i$, and $\mathbb{X}^u$ the points not selected from any location. The three types of points are shown in Fig. 2. We then expand formula 6 into point level considering these three types of points as

$$\mathbb{E}\{\text{Cos} < \nabla_X, \nabla'_X >\} = \sum_{i=0}^{N}[p_s \cdot \text{Cos}(\frac{\partial X_i^s}{\partial y_i}, \frac{\partial' X_i^s}{\partial' y_i})$$
$$+ p_g \cdot \text{Cos}(\frac{\partial X_i^g}{\partial y_i}, \frac{\partial' X_i^g}{\partial' y_i}) + p_u \cdot \text{Cos}(\frac{\partial X_i^u}{\partial y_i} \cdot \frac{\partial' X_i^u}{\partial' y_i})] \cdot \frac{1}{N}, \qquad (7)$$

where $N$ is the number of pixels in the output feature map. Denote $p_u$, $p_g$ and $p_s$ the probability of picking $X_i$ from $\mathbb{X}^u$ $\mathbb{X}^g$, and $\mathbb{X}^s$ in the second inference by the kernel from location $i$, respectively. It obvious that, $\text{Cos}(\frac{\partial X_i^u}{\partial y_i}, \frac{\partial' X_i^u}{\partial' y_i}) = 0$, $Cos(\frac{\partial X_i^s}{\partial y_i}, \frac{\partial' X_i^s}{\partial' y_i}) = 1$, and that $0 \leqslant \text{Cos}(\frac{\partial X_i^g}{\partial y_i}, \frac{\partial' X_i^g}{\partial' y_i}) \leqslant 1$. Taking into account the probabilities, for the fixed kernel,

$p_u$ and $p_g$ are both 0 and $p_s$ is 1. In comparison, our randomized kernel will decrease $p_s$ and increase $p_u$ and $p_g$. Thus, theoretically, the random kernel can reduce the attack's transferability in DCS.

Given Eq. 7, we then analyze how the hyperparameters of DCS affect the gradient similarity. Specifically, consider a DCS layer with a stride of $S$, initial kernel size of $k$, and receptive field of $n$. In pure random sampling, the probabilities in Eq. 5 can be represented as $p_s = \frac{n}{k}$, $p_g = P_g(n, k, S)$, and $p_u = 1 - p_s - p_g$, where $P_g(n, k, S)$ can be calculated via $n, k$ and $S$. The detailed constraints of $P_g(n, k, S)$ are given in the supplementary materials. $S$ and $k$ are limited by the dimension of the output feature map.

**Lemma 1.** *Consider a fixed stride and boundary of DCS as $S$ and $k$ separately. To guarantee the expectation of gradient cosine similarity to be smaller than a minimum amount $\epsilon_c$, the size of kernel $n$ is strictly upper bounded as*

$$n \leqslant S^2 \cdot \epsilon_c. \qquad (8)$$

*The detailed proof is in the supplementary material.*

To minimize the gradient similarity, $\epsilon_c$ should be minimized, which leads to minimizing $n$. In addition, since the variables in Eq. 8 are not related to data, the only undetermined hyperparameter of DCS $n$ can obtain a strict **data-independent** upper bound, which proved our claim of DCS to be data-independent in Sec. 4.1. This bound guarantees a small gradient similarity between each inference paths, thus improving the adversarial robustness of DCS.

### 4.3. Balancing Robust and Natural Accuracy

It has been proved that decreasing $n$ can minimize the gradient similarity. However, lowering the receptive field may lead to a reduction of robustness intuitively. So there is a trade-off between the natural accuracy and the gradient similarity. However, existing articles on stochastic defense usually fail to notice this trade-off. To bridge this limitation, in this section, we analyze this trade-off by deriving the relationship between $n$ and the natural accuracy. Intuitively, the receptive field in convolution will limit the performance of the convolution. To quantify the performance, we first assume an optimal path of DCS whose output feature map is $y$. Given the optimal case, we use the $L_1$ distances between $y$ and the output feature maps $y'$ from other inference paths to evaluate the performances of DCS on clean data. Specifically, under the definitions in Sec. 4.2, after multiplying normalization parameter $\nu$, the $L_1$ distance between the output feature maps, $\nu \cdot \|y - y'\|$ can be defined as

$$\nu \cdot \|y - y'\| = \sum_{i=0}^{N/S^2} | \sum_{K_j, K'_j \in \mathbf{K}, \mathbf{K}'} \nu_i[w(K_j) \cdot X_{j+i} \\ - \cdot w(K'_j) \cdot X'_{j+i}]|, \qquad (9)$$

**Algorithm 1** Gradient-Selective Adversarial Training

**Input:** Training dataset: $\mathbb{X}, \mathbb{Y}$; Classifier: $h(\mathbf{X})$ with initial weight: $w$ and repeated points $\mathbb{X}^s = \emptyset$; Attack iterations: $t$; Size of perturbation: $\epsilon_d$; Replaced layer numbers: $\mathbf{L}$; Size of deformed convolution kernel in each DCS: $n$; Other hyperparameters of each DCS: $S, k$

1: Sample $\delta$ from a set of *i.i.d.*standard Gaussian distribution;
2: Replace the convolution layer with DCS at layers $\mathbf{L}$ in the network;
3: **while** not converged **do**
4:     Sample a batch of $(\mathbf{X}, \mathbf{y}) \in (\mathbb{X}, \mathbb{Y})$;
5:     Initialize adversarial perturbation $\delta_x$;
6:     Sample a deformed convolution kernel for each DCS in the network;
7:     Updated and record the $\mathbb{X}^s$;
8:     **for** $i \leftarrow 1$ to $t$ **do**
9:         $\delta_x = clip_{\epsilon_d}(sign(\nabla_x \mathcal{L}(h(\mathbf{X}), \mathbf{y})) \cdot \eta)$;
10:     **end for**
11:     Set points in $\mathbb{X}^s$ as 0 and re-sample a deformed convolution kernel for each DCS in the network;
12:     $W = W - \nabla_W \mathcal{L}(h(\mathbf{X} + \delta_x,), \mathbf{y})$;
13: **end while**

---

**Algorithm 2** Adversarial Defense under White-Box Attack

**Input:** Test dataset: $\mathbb{X}, \mathbb{Y}$; Classifier: $h(\mathbf{X})$ with pretrained weight: $w$; Attack iterations: $t$; Size of perturbation: $\epsilon_d$; Replaced layer numbers: $\mathbf{L}$; Size of deformed convolution kernel in each DCS: $n$; Other hyperparameters in each DCS: $S, \kappa$

Replace the convolution layer with DCS at layers $\mathbf{L}$ in the network;

**Output:** $\tilde{\mathbf{y}}$

2: **while** having data not tested **do**
    Sample a batch of $(\mathbf{X}, \mathbf{y}) \in (\mathbb{X}, \mathbb{Y})$;
4:     Sample a new deformed convolution kernel for each DCS in the network;
    Initialize adversarial perturbation $\delta_x$;
6:     **for** $i \leftarrow 1$ to $t$ **do**
        $\Delta_x = clip_{\epsilon_d}(\eta \cdot sign(\nabla_x \mathcal{L}(h(\mathbf{X} + \delta_x), \mathbf{y})))$
8:     **end for**
    Re-Sample a new deformed convolution kernel for each DCS in the network;
10:     $\tilde{\mathbf{y}} = h(\mathbf{X} + \delta_x)$;
    **end while**

---

where $K$, $X$ versus $K'$, $X'$ represents variables in two different inference path, and $\nu_i$ is the normalization parameter at location $i$. Since $\mathbf{K}$ and $\mathbf{K}'$ are randomly sampled, the inconsistency varies at each sampling of DCS. Thus, we turn to analysis the expectation of the $L_1$ distance. To consider points in $\mathbb{X}^s, \mathbb{X}^g$ and $\mathbb{X}^u$ separately, we reformulate Eq. 9 by

$$\mathbb{E}\{\nu \cdot \|y - y'\|\} = \sum_{i=0}^{N/S^2} | \sum_{K_j, K'_j \in \mathbf{K}, \mathbf{K}'} p_u \nu_i [X'^u_{i+j} \cdot w(K'_j)]$$
$$+ p_g \nu_i [X^g_{i+j} \cdot w(K_{j-S*z}) - X'^g_{i+j} \cdot w(K'_j)] + p_s \cdot 0 |, \quad (10)$$

where $z$ is a drift that represents the number of sliding steps from $i$ to the location of the kernel that chooses points from $\mathbb{X}^g$. According to the derivation in Sec. 4.2 for $p_u$ and $p_g$, the inconsistency of the output feature map can be related to the DCS hyperparameter $n$.

**Lemma 2.** *Assume that the expectation of the inconsistency is less than a small number $\epsilon_l > 0$, the size of the deformed kernel is strictly lower bounded as,*

$$n \geqslant \frac{S^2 - 2kS + k^2 + k^4(1 - \frac{\epsilon_l S^2}{N})}{4kS} \quad (11)$$

*See detailed proof in the supplementary material.*

The lower bound on the definition of receptive field in Lemma 2 shows that to decrease the upper bound of the normalized distance, $n$ should be enlarged. In contrast, the upper bound in Lemma 1 wants to decrease $n$ to guarantee

a small gradient similarity between each inference path. To satisfy the trade-off between the gradient similarity and the natural accuracy, both bounds should be taken into account, which results in a data-independent constrain for $n$ as

$$\frac{S^2 - 2kS + k^2 + k^4(1 - \frac{\epsilon_l S^2}{N})}{4kS} \leqslant n \leqslant S^2 \cdot \epsilon_c. \quad (12)$$

With this constraint, we find the trade-off between attack transferability and natural accuracy while addressing the limitation of data dependency in randomized defense. In conclusion, when $n$ obeys the constraint in Eq. 12, DCS can trade-off between the adversarial transferability of different inference paths and the natural accuracy, and provide strong adversarial robustness to the applied network regardless of the challenges from any dataset.

### 4.4. Gradient-Selective Adversarial Training

The above theoretical derivation finds the trade-off in the data-independent case. In practice, however, Eq. 7 and 10 also remind us that data can affect trade-off. The points in $\mathbb{X}^g$ and $\mathbb{X}^u$ influence the gradient similarity and output inconsistency, while $X^s$ hardly affects these two values. To further explore the potential performance of the network on a specific dataset, we enhance the adversarial training by eliminating less significant cases corresponding to $X^s$. Specifically, we find $X^s$ by peeking the chosen inference paths of the attacks in AT, and then manually mask out (set 0) the points on the attacked paths. In formula, following

the definition of AT in Eq. 3, we have,

$$\mathcal{H}^* = \underset{h_2 \in \complement_{\mathcal{H}} h_1}{\arg\min} \mathbb{E}_{\mathbf{X},\mathbf{y} \sim \mathbb{X},\mathbb{Y}}[\mathcal{L}(h_2(\tilde{\mathbf{X}}_1'), \mathbf{y}], \qquad (13)$$

where $\complement_{\mathcal{H}} h_1 = h|h \in \mathcal{H}$, and $h \neq h_1$. The details of gradient-selective AT (GSAT) are shown in Algorithm 1.

During inference, we will not make any assumptions about the attack strategy to obtain a fair result. However, every DCS will sample $n$ points inside the boundaries purely randomly and independently as a new convolutional kernel for the DCS before each inference. The detailed inference method is shown in Algorithm 2.

## 5. Experiment

### 5.1. Experiment Setup

We evaluate DCS on CIFAR dataset (Krizhevsky, 2012) and Imagenet dataset (Krizhevsky et al., 2017). Various convolution-based networks are used for baseline networks, including ResNet18 (Park et al., 2022), ResNet50 (He et al., 2016) and WideResNet34 (Zagoruyko & Komodakis, 2016). In our experiments, unless specifically labeled, DCS replaces the second convolutional layer. All other layers keep the original settings.

**Experiments on CIFAR** The CIFAR 10 and CIFAR 100 datasets contain 10 and 100 classes, respectively. Each dataset consists $5.0 \times 10^4$ training samples and $1.0 \times 10^4$ test samples, with all images resized to 32x32 pixels with three color channels. We implement DCS by replacing the third convolutional layer on both ResNet18 and WRN34 for the CIFAR datasets. The hyperparameter $n$ in DCS is set to 2 and $k = 5$ for both networks. In our experiments, we split the dataset into batches of 128, setting the weight decay at $5.0 \times 10^{-4}$. We employed an SGD optimizer with a momentum of 0.9. The initial learning rate was set at 0.1 and was decreased according to a multi-step schedule. The model was trained over 200 epochs, with learning rate reductions by a factor of 10 at epochs 60 and 120. For adversarial training, we configured $\epsilon$ at $\frac{8}{255}$ and the step length $\eta$ at $\frac{2}{255}$ for a 7-step PGD (Madry et al., 2017). The model is implemented using PyTorch (Paszke et al., 2019) and trained on an NVIDIA GeForce RTX 4090 GPU.

**Experiments on ImageNet** The ImageNet dataset consists of $1.2 \times 10^6$ training examples and $5.0 \times 10^4$ test examples, classified into 1000 distinct classes. Each image is in a 3-channel RGB format with a resolution of $224 \times 224$ pixels. We implement DCS on ResNet50 for this dataset. The hyperparameters of DCS remain the same as on ResNet 18. In our experiments, we divide the dataset into batches of 512 examples. The weight decay is $1.0 \times 10^{-4}$ with the initial learning rate set at 0.1 and managed using a cosine annealing scheduler. The model is fine-tuned using adver-

sarial training for 90 epochs, with 10-step PGD (Madry et al., 2017) generating adversarial examples, setting $\epsilon$ to $\frac{4}{255}$ and the step length $\eta$ to $\frac{4}{255}$. The parameters for DCS are consistent with those from the CIFAR setup. The entire model is developed using PyTorch (Paszke et al., 2019) and evaluated using four NVIDIA GeForce RTX 4090 GPUs.

**Benchmarks** We evaluate DCS under SOTA attacks in TorchAttacks (Kim, 2020). For Projected Gradient Descent (PGD) (Madry et al., 2017), $\epsilon = 8/255$ with $2/255$ step size and 20 steps, employing random starts. We also assess Momentum Iterative Fast Gradient Sign Method (MIFGSM) (Dong et al., 2017) with $\epsilon = 8/255$, a step size of $2/255$, 5 steps, and a decay factor of 1.0. The DeepFool (Moosavi-Dezfooli et al., 2016) attack we used involves 50 steps with an overshoot of 0.02. We also perform the CW attack with a learning rate of 0.01, and AutoAttack (Croce & Hein, 2020) with $\epsilon = 8/255$. In addition, we evaluated DCS under complex attacks by Expectation over Transformations PGD (EOTPGD). All the experiments were repeated 10 times to get the mean value.

### 5.2. Main Results

We evaluated DCS on ResNet 18 and WideResNet 34 (WRN 34). The results are shown in Table 1. We compared DCS with other baselines to prove the advanced performance of DCS. In detail, we compare against a normal convolution layer and a normal DCN layer. Our DCS in main experiments performed GSAT whose effectiveness is proved later in ablation study in Sec. 5.3.2. As demonstrated in the tables, our approach significantly outperforms both standard convolution and fixed DCN. When applying DCS in ResNet 18 on the CIFAR 10 dataset, comparing against the normal network, our approach progressed 20.94% under the PGD attacks and 26.07% under AA, while making progress on natural accuracy of 7.94%. On CIFAR 100 dataset, the advancement of robust accuracy 18.26% under PGD attacks and 24.44% under AA, while progressing the natural accuracy by 9.57% On WRN 34 DCS with GSAT have also achieved high performances. In detail, on CIFAR 10, our approach is 21.84% more accurate under PGD attacks and 24.62% under AA. CIFAR 100 yields similar results at an advancement of 16.07% under the PGD attacks and 24.21% under AA. The natural accuracy also increases by 4.40% and 6.59% on CIFAR 10 and 100 respectively. From these results, DCS with GSAT are proven to have advanced performance on both CIFAR 10 and CIFAR 100.

### 5.3. Ablation Study

#### 5.3.1. INFLUENCE OF KERNEL SIZE ON ROBUSTNESS

As mentioned in Lemma 1, the initial kernel size $k$ of the deformed kernel can influence the transferability of the attacks

Table 1. The results of robust accuracy on CIFAR 10 and 100 of adversarial defense algorithms.

| Dataset | Model | Method | Natural | PGD$^{20}$ | AA | CW12 | MIFGSM | DeepFool |
|---|---|---|---|---|---|---|---|---|
| CIFAR 10 | RN 18 | Conv | 81.17 | 51.16 | 47.34 | 77.69 | 54.07 | 5.63 |
| | | DCN | 80.33 | 52.23 | 48.04 | 77.07 | 54.91 | 8.01 |
| | | DCS(ours) | **89.11** $\pm$ 0.10 | **72.10** $\pm$ 2.49 | **73.41** $\pm$ 2.04 | **88.56** $\pm$ 0.29 | **69.75** $\pm$ 2.05 | **88.24** $\pm$ 0.58 |
| | WRN 34 | Conv | 86.18 | 54.00 | 50.84 | 82.50 | 57.99 | 4.48 |
| | | DCN | 82.25 | 52.54 | 49.11 | 78.36 | 55.29 | 0.90 |
| | | DCS(ours) | **90.58** $\pm$ 0.08 | **75.84** $\pm$ 2.00 | **75.46** $\pm$ 1.62 | **90.28** $\pm$ 0.01 | **76.86** $\pm$ 1.79 | **90.20** $\pm$ 0.12 |
| CIFAR 100 | RN 18 | Conv | 55.14 | 28.31 | 24.57 | 50.61 | 29.55 | 5.07 |
| | | DCN | 54.06 | 23.23 | 19.82 | 48.88 | 25.41 | 1.05 |
| | | DCS(ours) | **64.71** $\pm$ 0.05 | **46.57** $\pm$ 1.96 | **49.01** $\pm$ 1.10 | **64.33** $\pm$ 0.07 | **49.38** $\pm$ 1.27 | **64.43** $\pm$ 0.26 |
| | WRN 34 | Conv | 60.10 | 31.98 | 28.15 | 55.35 | 33.87 | 3.47 |
| | | DCN | 58.90 | 27.02 | 24.47 | 53.88 | 29.65 | 4.79 |
| | | DCS(ours) | **66.69** $\pm$ 0.07 | **48.05** $\pm$ 1.81 | **52.36** $\pm$ 1.14 | **66.55** $\pm$ 0.06 | **51.24** $\pm$ 1.03 | **66.59** $\pm$ 0.30 |

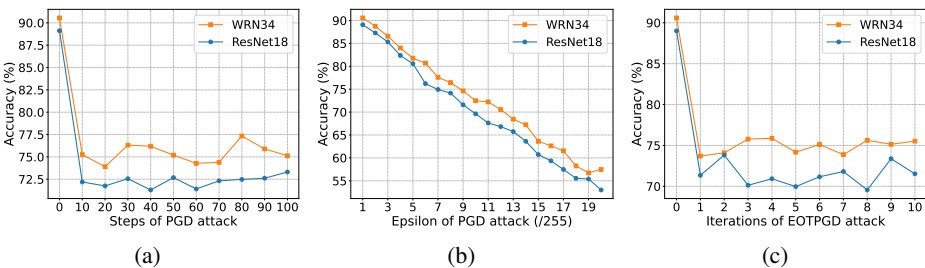

Figure 3. Evaluation of robustness under different attack settings: (a) PGD steps, (b) $\epsilon$ in PGD, (c) EOT iterations of EOTPGD attacks.

Table 2. Comparison with SOTA methods.

| Type | Methods | CIFAR 10 | | Imagenet | |
|---|---|---|---|---|---|
| | | PGD | AA | PGD | AA |
| AT | Overfit (Rice et al., 2020) | 55.06 | 52.24 | 39.85 | - |
| | DWQ (Fu et al., 2021) | 52.18 | 49.70 | 42.88 | - |
| | RobustWRN (Huang et al., 2021) | 59.13 | 52.48 | 31.14 | - |
| | WRN+DCN+AT (Zhu et al., 2018) | 52.54 | 49.11 | 40.66 | - |
| | DiffAT (Wang et al., 2023) | - | 70.69 | - | 31.30 |
| Random Noise | Additive Noise (Li et al., 2019) | 62.36 | 58.47 | - | - |
| | AdvWRN (Bartoldson et al., 2024) | - | 73.71 | - | - |
| | DF (Chen & Lee, 2024) | - | 58.22 | - | 40.60 |
| | MeanSparse (Amini et al., 2024) | - | 75.28 | - | 59.64 |
| Certified | Cert-RA (Chiang et al., 2020) | 68.60 | - | - | - |
| Random Structure | LWTA(Panousis et al., 2021) | **81.87** | 74.71 | - | - |
| | SAF (Wang et al., 2018) | 67.40 | - | - | - |
| | DCS(ours) | 75.84 | **75.46** | **52.38** | **66.79** |

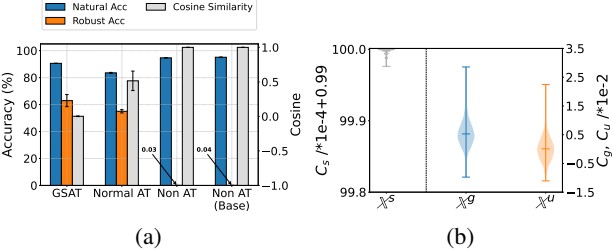

Figure 4. Observation of Cosine similarity: (a) comparison between different training techniques, (b)comparison between kernels that sample points from different collections.

Table 3. Evaluation on Gradient-Selective AT.

| Baseline | AT method | Natural Acc | Robust Acc |
|---|---|---|---|
| ResNet 18 | Normal AT | 83.53 $\pm$ 0.35 | 60.64 $\pm$ 0.02 |
| | GSAT | 89.11 $\pm$ 0.10 | 73.41 $\pm$ 2.04 |
| WRN 34 | Normal AT | 90.58 $\pm$ 0.10 | 54.45 $\pm$ 0.00 |
| | GSAT | 90.58 $\pm$ 0.08 | 75.46 $\pm$ 1.62 |

by controlling the upper bound of $n$. We evaluate the impact of the initial kernel size on the robustness of DCS under $PGD^{20}$ attacks, given a fixed receptive field. The results are shown in Table. 4. Specifically, we fixed the receptive field to 4 and then tested the case with initial kernel sizes of

3,5,7 and 9. The results show that larger boundaries help to enhance robustness and also provide more stable results. However, changing $k$ changes the padding and introduces more network parameters. Therefore, $k$ is recommended to be small. In the table, the robustness increases from $k = 3$ to 5. With $k > 5$, more additional parameters will be introduced, but the increases are smaller.

### 5.3.2. EVALUATION ON GRADIENT-SELECTIVE AT

The AT that is adapted to DCS is expected to boost the robustness via decreasing the attack transferability, as mentioned in Sec. 4.4. We believe that this adversarial training can effectively minimize the effect of the batch of points

Table 4. Effectiveness of DCS over different $k$ w/o GSAT.

| $k$ | Natural Acc | Robust Acc |
|-----|-------------|------------|
| 9*9 | $87.12 \pm 0.12$ | $\mathbf{68.64 \pm 1.14}$ |
| 7*7 | $\mathbf{90.63 \pm 0.10}$ | $66.39 \pm 3.20$ |
| 5*5 | $90.58 \pm 0.10$ | $62.93 \pm 4.48$ |
| 3*3 | $15.44 \pm 0.27$ | $37.52 \pm 5.29$ |

Table 5. GSAT over different receptive fields with a normal test against the test that canceled $\mathbb{X}^u$.

| $n$ | Include $\mathbb{X}^u$ | $\mathrm{PGD}^{20}$ | AA |
|-----|-----------------------|---------------------|-----|
| 4 | ✔ | $62.93 \pm 4.48$ | $64.18 \pm 0.91$ |
| 3 | ✔ | $70.75 \pm 2.55$ | $72.88 \pm 1.47$ |
| 2 | ✔ | $\mathbf{72.10 \pm 2.49}$ | $\mathbf{73.41 \pm 2.04}$ |
| 1 | ✔ | $61.83 \pm 1.25$ | $64.40 \pm 1.89$ |
| 4 | ✘ | $63.50 \pm 3.80$ | $56.40 \pm 1.53$ |
| 3 | ✘ | $\mathbf{65.15 \pm 3.26}$ | $65.83 \pm 0.88$ |
| 2 | ✘ | $64.64 \pm 2.04$ | $\mathbf{66.14 \pm 1.60}$ |
| 1 | ✘ | $56.82 \pm 1.84$ | $58.88 \pm 1.67$ |

$X^s$ on the network and decrease the gradient similarity. To measure its effectiveness, we first observe the gradient similarities and accuracies of ResNet 18 using GSAT, AT, and normal training on CIFAR 10. The results are shown in Fig. 4a. It can be found that the GSAT reduces the cosine similarity and boosts the natural accuracy compared to AT, which consists of our expectations. For more generalized situations, We evaluated the GSAT on CIFAR 10 and 100 using ResNet 18 and WRN 34 as the baselines under AutoAttack. The results are shown in Table. 3. We can find that with larger $n$, GSAT will be more unstable. We give an explanation as: when Eq. 12 gives **larger bounds** for $n$, which refers to larger $n$ in practice, the percentage paths in unselected random space will increase. This increases the probability for DCS performance decrement in the networks using GSAT for training, which leads to greater instability. Thus, as shown in Table. 3, $n$ is suggested to be small for a stable GSAT. Table. 3 also shows that DCS obtains higher robust accuracy and natural accuracy via GSAT, which is consistent with our analysis.

### 5.3.3. OBSERVATION OF GRADIENT SIMILARITY

To further verify the validity of the study of Eq. 7 on gradient similarity, we directly observed the three cases corresponding to $\mathbb{X}^s$, $\mathbb{X}^g$ and $\mathbb{X}^u$. Specifically, we manually designed three sets of deformed kernels with $n = 4$ corresponding to the three cases represented. The detailed kernel is illustrated in the supplementary material. We used ResNet 18 as a baseline and randomly selected 1000 points on CIFAR 10 for observation, and the distribution of gradient similarity for each case is shown in Fig. 4b. In this figure $C_s$ is $\mathrm{Cos}\left(\frac{\partial X^s}{\partial y_i}, \frac{\partial' X^s}{\partial' y_i}\right)$, $C_g$ means $\mathrm{Cos}\left(\frac{\partial X^g}{\partial y_i}, \frac{\partial' X^g}{\partial' y_i}\right)$, $C_u$

means $\mathrm{Cos}\left(\frac{\partial X^u}{\partial y_i}, \frac{\partial' X^u}{\partial' y_i}\right)$. The results are consistent with what we analyzed in Sec.4.2, proving that the analysis in Lemma 1 on Eq. 7 and the data-independent upper bound of $n$ are consistent with the facts. Moreover, according to our observation, the cosine similarity is also low when points belonging to $\mathbb{X}^g$ are sampled the second time. We hypothesize that this is due to the different sources of gradients generated by the two inference paths for the same point. This phenomenon also implies that the upper bound of $n$ can be loosened a bit in some datasets.

### 5.3.4. INFLUENCES ON RECEPTIVE FIELD

To verify the trade-off mentioned in Sec. 4.3, we evaluated the performance of DCS over different receptive fields. Specifically, we use ResNet 18 as the baseline for this ablation study and follow the settings of the major experiments. Given the observation about the gradient similarity on CIFAR 10, as analyzed in Sec. 5.3.3, the upper bound of $n$ can be loosened. Therefore, we evaluate of the effect of $n$ in a wide range of values on CIFAR 10. The results are shown in the first four lines of Table. 5, where DCS-applied networks achieved the highest robust accuracy of $72.10\%$ under PGD attacks and $73.41\%$ under AA with the receptive field of 2. From the table, there is a clear trend of trade-off for $n$ on robustness, which consists of our derivation in Eq. 12.

Furthermore, based on the observation in Sec. 5.3.3 about the cosine similarity of points in $\mathbb{X}^g$, if Eq. 7 holds, then the DCS will maintain strong robustness even if $p_u = 0$. To verify the validity of Eq. 7, we manually fix $p_u = 0$ by canceling points in $\mathbb{X}^u$ in the second inference. The results are shown in the last four lines in Table. 5. The best robustness is higher than the baseline but suffers a minor decrement compared to the DCS including points in $\mathbb{X}^u$. This result proves that in Eq. 7, $0 < \frac{\partial X_i}{\partial X} \cdot \frac{\partial' X_i}{\partial' X^g} < 1$. In addition, given our observations on gradient similarity, Eq. 7 holds, which verifies the validity of Lemma 1.

### 5.3.5. ROBUSTNESS UNDER DIFFERENT STRENGTH OF ATTACK

We evaluated the defense capacity of DCS against attacks of different strengths using ResNet 18 and WRN 34 as the baseline on CIFAR 10. We evaluate DCS via three groups of experiments under different settings of PGD attacks.

**PGD steps.** Since the effectiveness of white-box attacks such as PGD depends on strength and adversarial transferability. In order to avoid the effect of attack strength on the robustness provided by the DCS, we conducted experiments on the CIFAR 10 dataset using ResNet 18 and WRN 34 with DCS under PGD attacks of different strengths. The evaluations are performed at different levels of PGD attacks at each 10 step from clean settings, until 100 steps. The

Table 6. DCS under black-box attacks.

| Baseline | Natural | Query-based | | Transfer-based | | Adaptive | |
|---|---|---|---|---|---|---|---|
| | | SQUARE | Pixel | FGSM | FGSM-base | BPDA | BPDA+EOT |
| RN18 | 89.11 | 80.53 | 86.32 | 84.95 | 65.71 | 78.66 | 77.78 |
| WRN34 | 90.58 | 82.45 | 86.31 | 90.53 | 68.34 | 80.03 | 80.47 |

Table 7. DCS applied to ViT.

| Method | PGD |
|---|---|
| ViT-t+Conv | 32.31 |
| ViT-t+DCS | **55.71** |

results are shown in Fig. 3a. We found that even under strong attacks, PGD can still provide strong robustness to the baseline network, despite some fluctuation.

**PGD perturbation boundary $\epsilon$.** In addition to the changed number of steps for generating the adversarial perturbation, the attack strength can be changed by varying the boundaries of the adversarial perturbation $\epsilon_d$. We evaluated the performance of DCS under attacks of different $\epsilon_d$. The results are shown in Fig. 3b. It can be seen that the robustness of the DCS is gradually decreasing as the perturbations become stronger. This suggests that DCS has difficulty in reducing the adversarial transferability of an attack to 0. We attribute this to the fact that the trade-off prevents the cosine similarity from being reduced close to 0.

**Iterations of EOTPGD** Finally, we also evaluate the robustness of the DCS under multi-step attacks. Specifically, we used EOTPGD as a benchmark for multi-step attacks and evaluated the robustness of models embedded with DCS under EOT attacks from 1 to 10 EOT steps. The results of which are shown in Fig. 3c, which shows that DCS remains robust even under multi-step EOTPGD attacks. This indicates that DCS can adapt to complex multi-step attacks.

### 5.3.6. ROBUSTNESS UNDER BLACK-BOX ATTACKS

Considering from the gradient perspective, DCS objectively shields a portion of the gradient. The defense performance of DCS, as a gradient-masking defense method, against black-box attacks is difficult to theoretically infer.

To have a complete understanding of the robustness of DCS, we observed the defense capability of DCS against black-box attacks. We evaluate DCS under query-based, transfer-based (Goodfellow et al., 2014b) and adaptive attacks (Athalye et al., 2018a) on CIFAR 10 with ResNet 18 and WideResNet 34. The results are shown in table 6. DCS is found to have some defense capability against black-box attacks. We attribute this phenomenon to the fact that the random masking of the convolutional kernel by DCS also shields a portion of the adversarial perturbations, which de-

stroys the integrity of the black-box attack thereby reducing its effectiveness. For ataptive attacks like BPDA and EOT-BPDA, DCS also shows high robustness. We attribute this to randomness in conjunction with gradient masking.

### 5.3.7. DCS ON TRANSFORMER

We confirm that DCS is tailor-made for convolutional operations. We believe that convolution is still an important tool in image processing. Transformer is out of the research scope of this work. However, we briefly explored that DCS fits Vision Transformer (ViT).

ViT uses a $16 \times 16$ convolution in patch embedding. Large kernel size makes lower bound of $n$ increases and hinders finding a suitable $n$. To avoid this, we notice that patch embedding can be split into multiple concatenated $3 \times 3$ convolutions. Our baseline follows the settings in (Xiao et al., 2021) and then replace the second $3 \times 3$ convolution with DCS. The results are shown in Table 7. We can find that DCS also achieves strong robustness on ViT.

**Training Stages.** The network is trained on CIFAR-10 in two stages. All stages are trained using SGD as the optimizer, with a weight decay of 5e-4, an initial learning rate of 0.01 and a batch size of 128. Stage one fixes a pretrained ViT-tiny and trains the FC layer and convolutions from scratch with 200 epochs using the multistep scheduler, where the learning rate is divided by 10 at epoch 50 and 100. Stage two adversarially finetunes the entire network using GSAT with 90 epochs with a cosine scheduler.

## 6. Conclusion

In this paper, we proposed a random structural defense method named DCS. We proved that DCS solved two major limitations in existing random defense methods, which are data-dependent hyperparameters and the trade-off between gradient similarity and natural accuracy. Through theoretical derivations, we design a set of bounds for DCS to reduce the adversarial transferability between each of its inference paths without degrading the natural accuracy. Finally, we design GSAT for DCS to enhance the robustness of DCS. We experimentally validate the above theory and demonstrate the SOTA robustness of DCS. In future research, we expect that data-independent defense methods similar to DCS can demonstrate defenses in various areas of computer vision to fit 0-shot inference scenarios.

## Acknowledgement

This work was supported in part by the Start-up Grant (No. 9610680) of the City University of Hong Kong, Young Scientist Fund (No. 62406265) of NSFC, and the Australian Research Council under Projects DP240101848 and FT230100549.

## Impact Statement

This paper presents work whose goal is to advance the field of Machine Learning. There are many potential societal consequences of our work, none of which we feel must be specifically highlighted here.

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

# A. Proof of Theorems

## A.1. Proof of Lemma 1

We start with explaining the pixel-level gradient similarity. The cosine similarity in pixel level can be redefined as,

$$\text{Cos} < \nabla_X, \nabla'_X >= \frac{\sum_{i=0}^{N} \nabla_{Xi} \nabla'_{Xi}}{M}, \tag{14}$$

where $M = \|\nabla_X\|\|\nabla_X\|$. For each poin, considering the three types, the cosine similarities can be divided into three terms as is demonstrated in Eq. 7 in the main text. We define $\mathbb{E}(\text{Cos} < \nabla_{Xi} \nabla'_{Xi} >) = \sum_{i=0}^{N} C_i \frac{\|\nabla_{Xi}\|\|\nabla'_{Xi}\|}{M}$, where $C_i = \text{Cos} < \nabla_{Xi}, \nabla'_{Xi} >$ represents the gradient similarity at each poin. Considering $p_g = 1 - p_u - p_s$, we have

$$C_i = \frac{n}{k^2}(1 - C_i^g) + C_i^g - p_u \cdot C_i^g, \tag{15}$$

where $C_i^g = \frac{\partial X_i^g}{\partial y_i} \cdot \frac{\partial' X_i^g}{\partial' y_i} \in [-1, 1]$. When $C_i^g \leqslant 0$, it can be guaranteed that $C_i \leqslant \frac{n}{k^2}$. Thus, we concentrate on the case that $C_i^g > 0$. From Eq. 15, $C_i^g$ is decided by the data and the network parameters. As the data is unknown, to minimize $C_i$, we maximize $p_u$. Since $p_u$ is decided by the random sampling results, we turn to maximize the lower bound of $p_u$ to guarantee a small upper bound of $C_i$. Given a point $X_0$ in the first inference, the $\mathbb{X}^g$ produced by it can stay in $S^2$ different shapes. There are three kinds of sizes for those differently shaped $\mathbb{X}^g$. Denote $m \equiv k^2 \pmod{S^2}$, the number of $\mathbb{X}^g$ in size $\lceil \frac{k}{S} \rceil^2$ is $m^2$, in size $\lceil \frac{k}{S} \rceil \lfloor \frac{k}{S} \rfloor$ is $2m(k-m)$, and in size $\lfloor \frac{k}{S} \rfloor^2$ is $(k-m)^2$, where $\lfloor \cdot \rfloor$ for rounding down and $\lceil \cdot \rceil$ for rounding up. Denote $\frac{k}{S} = \alpha$, and $(k-m) = \beta$, the lower bound of $p_u$ in Eq. 7 in the main text can be defined as

$$p_u \geqslant \begin{cases} \frac{k^2 - n\lceil\alpha\rceil^2}{k^2}, & \text{if } n \leqslant m^2; \\ \frac{k^2 - m^2\lceil\alpha\rceil^2 - (n-m^2)\lceil\alpha\rceil\lfloor\alpha\rfloor}{k^2}, & \text{if } m^2 < n \leqslant m^2 + 2m\beta; \\ \frac{(S^2 - n)\lfloor\alpha\rfloor^2}{k^2}, & \text{if } m^2 + 2m\beta < n \leqslant S^2; \\ 0, & \text{if } n > S^2, \end{cases} \tag{16}$$

From Eq. 17, when $n > S^2$, the lower bound of $p_u$ is 0 which makes no guarantee that the $C_i$ is small. Thus we concentrate on the cases where $n < S^2$. When $n < S^2$, a looser bound of $p_u$ from Eq. 16 can be derived as

$$p_u \geqslant \frac{(S^2 - n)\lfloor\alpha\rfloor^2}{k^2}. \tag{17}$$

Then, as the upper bound of $C_i$ is expected to be smaller than $\epsilon_c$, it can be formulated as

$$C_i \leqslant \frac{n}{k^2}(1 - C_i^g) + C_i^g - \frac{(S^2 - n)\lfloor\alpha\rfloor^2}{k^2} \cdot C_i^g \leqslant \epsilon_c, \tag{18}$$

From Eq. 18, the upper bound of $n$ can be derived as

$$n \leqslant \frac{k^2\epsilon_c + (S^2\lfloor\alpha\rfloor^2 - k^2)C_i^g}{1 + (\lfloor\alpha\rfloor^2 - 1)C_i^g}. \tag{19}$$

When considering the worst case, where $C_i^g = 1$, the upper bound of $n$ can be rewritten as

$$n \leqslant \frac{k^2(\epsilon_c - 1)}{\lfloor\alpha\rfloor^2} + S^2. \tag{20}$$

From Eq. 20, as $\lfloor\alpha\rfloor \leqslant \frac{k^2}{S^2}$, there is a looser bound for $n$ as

$$n \leqslant S^2\epsilon_c. \tag{21}$$

## A.2. Proof of Lemma 2

In Lemma. 2, we constrain the $L_1$ distance between the output features after normalization into a small boundary. To expand the $L_1$ distance into pixel level, we divide the points into three type, which are sampled from $\mathbb{X}^s$, $\mathbb{X}^g$, and $\mathbb{X}^u$ respectively,

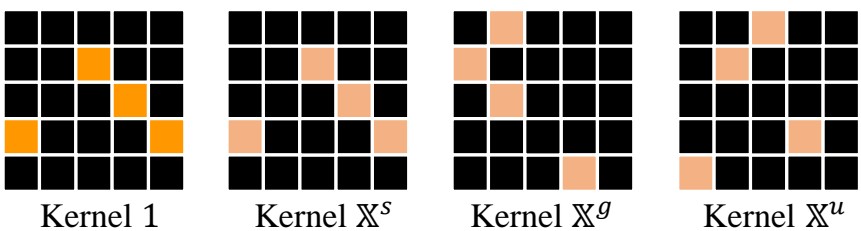

*Figure 5.* Illustration of Kernel Pairs

as shown in Eq. 10 in the main text. Denote $\overline{\Delta^g} \in [0, 1]$ by the average value of $\sum_{j=0}^{k^2} \nu_i[X_{i+j}w(K_j) - X'^g_{i+j}w(K'_j)]$ and $\|\gamma\|$ by the expectation of $\sum_{i=0}^{N/S^2} \nu_i \sum_{j=0}^{k^2} [X]_{iS^2+j}w(K'_j)$[1]. the constrain in $L_1$ distance in pixel level is

$$p_u\|\gamma\| + \frac{(1 - p_u - \frac{n}{k^2})\overline{\Delta^g}N}{S^2} \leqslant \epsilon_l \tag{22}$$

To satisfy Eq. 22 we compare the boundary with the upper bound of $p_u$. Similar to the analysis in Section. A.1, we divide the upper bound of $p_u$ into three cases as

$$p_u \leqslant \begin{cases} \frac{k^2 - \lceil\frac{n}{\lceil\alpha\rceil^2}\rceil\lceil\alpha\rceil^2}{k^2}, & \text{if } n \leqslant m^2\lceil\alpha\rceil^2 \\ \frac{k^2 - m^2\lceil\alpha\rceil^2 - \lceil\frac{(n-m^2)}{\lceil\alpha\rceil\lfloor\alpha\rfloor}\rceil\lceil\alpha\rceil\lfloor\alpha\rfloor}{k^2}, & \text{if } m^2\lceil\alpha\rceil^2 < n \leqslant k^2 - \beta^2\lfloor\alpha\rfloor^2; \\ \frac{\lfloor\frac{(k^2-n)}{\lfloor\alpha\rfloor^2}\rfloor\lfloor\alpha\rfloor^2}{k^2}, & \text{if } k^2 - \beta^2\lfloor\alpha\rfloor^2 < n; \end{cases} \tag{23}$$

For the 3 terms to the right hand side in Eq. 23, the goal is to leave only $n$ into the numerator part. Thus the relationship between $p_u$ and $n$ can be clear. We loose the terms by considering $-\lceil\cdot\rceil$ as $-(\cdot)$ and $\lfloor\cdot\rfloor$ as $(\cdot)$ For the first term

$$pu \leqslant 1 - \frac{n\alpha^2}{\lceil\alpha\rceil^2 k^2}$$

For the second term

$$p_u \leqslant 1 - (\frac{m^2\alpha^2}{k^2} + \frac{n\alpha}{k^2\lceil\alpha\rceil} - \frac{m^2\alpha}{k^2\lfloor\alpha\rfloor}) \leqslant 1 - (\frac{n\alpha}{k^2\lceil\alpha\rceil})$$

For the third term

$$p_u \leqslant 1 - \frac{n\alpha^2}{k^2\lfloor\alpha\rfloor^2}$$

We set the strictest bounds for $p_u$. Bringing in $\alpha = \frac{k}{S}$, there is a looser boundary of $p_u$ considering all cases as

$$p_u \leqslant 1 - \frac{n}{\lfloor\alpha\rfloor^2 S^2}, \tag{24}$$

where $n < \lfloor\alpha\rfloor^2 S^2$. As mentioned in Section. A.1, we consider the situations where $n < S^2$, and $\lfloor\alpha\rfloor \geqslant 1$, $n$ always satisfies the condition $n < \lfloor\alpha\rfloor^2 S^2$. To satisfy Eq. 22, for given $k$ and $S$, $n$ must be greater than a lower bound as

$$n \geqslant \frac{\lfloor\alpha\rfloor^2 - \lceil\alpha\rceil^2(\frac{\epsilon_l S^2 k^2}{N\overline{\Delta^g}} - 1)k^2}{\lceil\alpha\rceil^2 - \lfloor\alpha\rfloor^2} \tag{25}$$

As $\alpha - 1 \leqslant \lfloor\alpha\rfloor \leqslant \alpha$ and $\alpha \leqslant \lceil\alpha\rceil \leqslant \alpha + 1$, a looser boundary can be found for $n$ as

$$n \geqslant \frac{(\alpha - 1)^2 - \alpha^2 k^2(\frac{\epsilon_l S^2}{N\overline{\Delta^g}} - 1)}{4\alpha} \tag{26}$$

---

[1]$N$ here is the size of input feature map including the padding point

When consider the worst case where $\overline{\Delta^g} = 1$, taking $\alpha = \frac{k}{S}$ into Eq. 26, $n$ is lower bounded as

$$n \geqslant \frac{S^2 - 2kS + k^2 - k^4(\frac{\epsilon_l S^2}{N} - 1)}{4kS}. \tag{27}$$

The form in Eq. 27 consists of the form in Lemma. 2.

### A.3. Constrain of $P_g$

Based on the boundary of $p_u$ mentioned in Section. A.1 and Section. A.2, we can derive the boundary of $P_g$ from $P_g = 1 - p_u - p_s$ as

$$\frac{n}{\lfloor \alpha \rfloor^2 S^2} - \frac{n}{k^2} \leqslant P_g \leqslant 1 - \frac{(S^2 - n)\lfloor \alpha \rfloor^2 - n}{k^2}, \tag{28}$$

where $n < S^2$. From Section. A.1 and Section. A.2, we claim that in the derivation of Lemma.1 and Lemma. 2, $P_g$ is bypassed through replacing $P_g$ into $1 - p_u - p_s$, where $p_s = \frac{n}{k}$. The exact form of $P_g$ is not important for Lemma. 1 and Lemma. 2. Thus we do not specify $P_g$ in the derivation in Sec. 4.

## B. Additional Experiments

### B.1. Explanation of the instability of GSAT

According to Eq. 13, GSAT modifies the sample space sampled by the DCS layer in forward propagation during training by selection. Below we explain the reason for the instability of GSAT shown in Table 3.

We have demonstrated both theoretically and experimentally that paths in $\mathbb{X}^g$ and $\mathbb{X}^u$ are not easy to attack, while paths in $\mathbb{X}^s$ are very easy to be attacked by gradient-based algorithms (see Eq. 7, 10 and Sec. 5.3.3 for details). This means that when the network is attacked, the gradients generated by the paths in $\mathbb{X}^g$ and $\mathbb{X}^u$ will be in different direction from the attacked gradients generated by the paths in $\mathbb{X}^s$.

To minimize the influence of inaccurate attacked gradients in training, we select and remove paths in $\mathbb{X}^s$ from $\mathbb{X}$, and build the selected random space, as demonstrated in Eq. 13. This corresponds to step 11 in Algorithm 1. The selected random space is used for sampling paths in the forward and backward propagation for each training step separately.

We understand that using GSAT the network are only limited optimized on the paths in $\mathbb{X}^s$ under each adversarial examples. So when the DCS samples paths in $\mathbb{X}^s$ in inference, there are performance decrement compared to the other sampled paths. The decrement will cause the instability in GSAT. The larger probability the performance drop in DCS, the greater instability in GSAT.

### B.2. Illustration of Toy Kernel Pairs

To observe the gradient similarity in Section 5.3.3 in the main text, we choose some toy kernel pairs, where the points sampled in the second kernels are in $\mathbb{X}^s$, $\mathbb{X}^g$, and $\mathbb{X}^u$ respectively. The shapes of these toy kernel pairs are illustrated in Fig. 5. In the figure, Kernel 1 represents the kernel sampled in the first inference, and Kernel $\mathbb{X}^s$, $\mathbb{X}^g$, and $\mathbb{X}^u$ represents the kernel sampled with points only in collection $\mathbb{X}^s$, $\mathbb{X}^g$, and $\mathbb{X}^u$, respectively.

### B.3. Grid Search of $k$ and $n$ in DCS

We did grid search on the hyperparameters of DCS. As $S$ is determined by the network, there lefts two hyperparameters to be determined, which are $k$ and $n$. The results are shown in Fig. 6. From the figure, it is clear that under the assumption of $n < S^2$, when $k = 5$ and $n = 2$, DCS can reach the best performance under both AA and PGD attacks. Thus, we choose this group of hyperparameters in the major experiments.

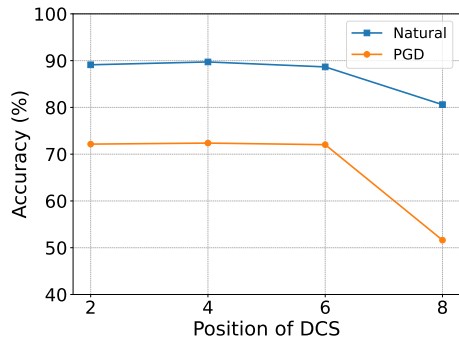

Figure 7. DCS at Different Position

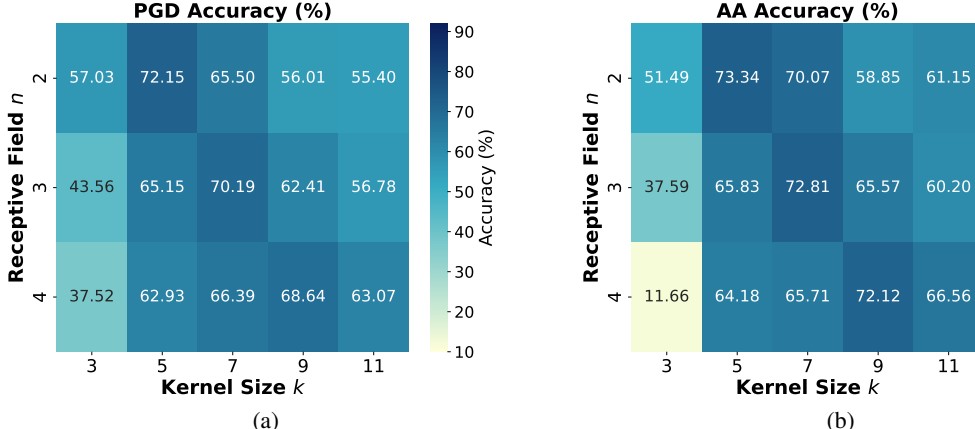

*Figure 6.* Heatmaps of accuracy under different attack settings: (a) PGD, (b) AA.

## B.4. DCS at Different Position

We initially explored the accuracy of DCS when implemented at various positions within a ResNet under PGD[20] attacks, the results of which are illustrated in Fig. 7. Our findings indicate that the accuracy is relatively close to a peak value in the first six layers. The accuracy at the eighth layer is comparatively lower than at other positions. Consequently, in our major experiments, we choose the second convolution layer to be replaced with the DCS layer.

## B.5. Computational Overhead

Despite the difference in training time, their time consumption during inference is very similar. The time gap in training is due to the introduction of additional parameters in the DCS layer. However, during inference, the extra time consumption caused by the extra parameters is not as dramatic as training due to the absence of backpropagation. The results are shown in Table 8.

*Table 8.* Training and reasoning overhead comparison.

| Model | Training Overhead/min | Reasoning Overhead/sec |
|---|---|---|
| baseline | 213.47 | 5.94 |
| DCS | 379.96 | 6.67 |

## B.6. Impact of Downsampling Layer Substitution

We compare the results by replacing the DCS to the downsampled layer (layer 6) with the closest normal convolution layer (layer 5, 7). The results are shown in Table 9.

*Table 9.* Downsampling layer substitution.

| Layer | PGD | AA |
|---|---|---|
| 5 | 68.12 | 70.72 |
| 6 | **72.03** | **75.74** |
| 7 | 69.36 | 72.11 |

We notice that the performance of DCS on downsampling layers is better than the normal convolution layers. According to the lemmas, larger stride $\mathcal{S}$ helps to reduce the assumed small numbers $\epsilon_c$ and $\epsilon_l$ in the lemmas. It results in a smaller gradient similarity and output distance, and finally increases the robustness and clean accuracy. In addition, larger $\mathcal{S}$ also helps the bounds in both lemmas to be numerically looser in practice. This helps to find a suitable $n$ easier.

