# OpenReview forum: "Adversarial Robustness via Deformable Convolution with Stochasticity"
_ICML.cc/2025/Conference — ICML 2025 poster_

### Official Review · Reviewer_7KWx · 2025-03-11

**Overall Recommendation:** 4

**Summary:**

This paper introduces DCS (Defensive Convolution with Stochasticity), a novel adversarial defense method that integrates randomness directly into convolution operations to obscure gradient directions. By embedding stochasticity within the network architecture, DCS enhances robustness against both white-box and black-box attacks. The authors provide theoretical analysis and experimental validation across multiple datasets and adversarial settings.

**Claims And Evidence:**

The paper claims that DCS effectively mitigates gradient-based attacks, does not require post-training modifications, generalizes well across architectures, and is data-independent. These claims are supported through theoretical derivations, empirical results, and ablation studies. However, the notion of "data independence" needs clarification, as it appears to apply to hyperparameters rather than the full training process.

**Essential References Not Discussed:**

none

**Experimental Designs Or Analyses:**

The empirical results effectively demonstrate the effectiveness of DCS. However, terminology could be improved—using "pixel" to describe both feature maps and convolution kernels may cause confusion. Figure 1 also needs more explanation, especially regarding the meaning of the circles between Step 3 and Step 4.

**Methods And Evaluation Criteria:**

The experiments are well-designed, covering a diverse range of attacks, datasets, and architectures. Evaluation metrics focus on robust accuracy, standard accuracy, and computational complexity. The approach is rigorous, but clearer documentation of the training process and hyperparameter choices would improve reproducibility.

**Other Comments Or Suggestions:**

Clarifying the training process and improving the explanation of key theoretical components would enhance the paper’s impact. More precise terminology and a clearer interpretation of Figure 1 would also help avoid confusion.

**Other Strengths And Weaknesses:**

A major strength is the seamless integration of adversarial defense within convolution operations, eliminating the need for post-hoc modifications. The evaluation is comprehensive, and the theoretical contributions are meaningful. However, issues with terminology, theoretical clarity, and figure explanations should be addressed.

**Questions For Authors:**

Is Step 4 trained alternately or in a unified manner? Do the circles between Step 3 and Step 4 in Figure 1 represent an expansion of the potential distribution range? What aspect of the method is truly "data-independent"?

How does DCS compare with other stochastic adversarial defenses?

Would combining DCS with adversarial training further improve robustness?

**Relation To Broader Scientific Literature:**

The paper situates itself well within adversarial defense research but could better contrast DCS with other stochastic defenses, such as randomized smoothing or stochastic activation functions. Citing additional references on these topics would strengthen the discussion.

**Theoretical Claims:**

The theoretical foundation is strong, but some areas need further elaboration. Lemma 2, in particular, lacks sufficient explanation, and its derivation should be expanded. Mathematical notation could also be refined to enhance clarity.

---

> ### Author Rebuttal · Authors · 2025-03-31
>
> Thank you for your detialed comments and your interest in the content of our experiments. We summarize and rebut your 8 major concerns in your comments.
>
> ## Re 0. The notion of "data independence" needs clarification.[Claims,Q3]
> Thank you for correcting our statement. The "data independence" means the hyperparameters at the DCS layer are data independent. However, the DCS layer still needs to be trained. We will revise it in the final version.
>
> ## Re 1. Clearer training process and hyperparameter choices.[Evaluation]
> Thank you for this nice concern. The hyperparameters of the experiments for all baselines and datasets are listed separately in the Sec. 5.1. In terms of the locations of replaced DCS layer, we conduct an ablation study on DCS locations in Sec. B.3, where we changed the location of DCS and find that the second layer is the best for DCS. For clarification, we will add this setting to Sec. 5.1. in the final version as:
>
> - In our experiments, unless specifically labeled, DCS replaces the second convolutional layer. All other layers keep the original settings.
>
> ## Re 2. Explanation of Lemma 2.[Theorem,Supplementary]
> Thank you for your advice. We noticed 2 major concerns for the explanation of Lemma 2, we will improve them in the appendix as follows.
>
> **(1)** We note that there is no explanation for the symbols $\lfloor\cdot\rfloor$ and $\lceil\cdot\rceil$. This will be explained in line 570 when it first appears as:
>
> - $\lfloor\cdot\rfloor$ for rounding down and $\lceil\cdot\rceil$ for rounding up.
>
> **(2)** Regarding the proof of Lemma 2, we note that the step from Eq. (23) to Eq. (24) need to be expanded:
>
> For the 3 terms to the right hand side in Eq. 23, the goal is to leave only $n$ into the numerator part. Thus the relationship between $p_u$ and $n$ can be clear. We loose the terms by considering $-\lceil\cdot\rceil$ as $-(\cdot)$ and $\lfloor\cdot\rfloor$ as $(\cdot)$
>
> - For the first term $$pu \leqslant 1-\frac{n\alpha^2}{\lceil\alpha\lceil^2k^2}$$
> - For the second term $$p_u\leqslant 1-(\frac{m^2\alpha^2}{k^2}+\frac{n\alpha}{k^2\lceil\alpha\rceil}-\frac{m^2\alpha}{k^2\lfloor\alpha\rfloor})\leqslant 1-(\frac{n\alpha}{k^2\lceil\alpha\rceil})$$
> - For the third term $$p_u\leqslant 1-\frac{n\alpha^2}{k^2\lfloor\alpha\rfloor^2}$$
> - We set the strictest bounds for $p_u$. Bringing in $\alpha=\frac{k}{S}$, the third term is found to be the strictest: $1-\frac{n}{S^2\lfloor\alpha\rfloor^2}$, as is shown in Eq. 24.
>
> After bringing the bounds in Eq. 24 into Eq. 22 and simplify it, Eq. 25 can be obtained. After transforming the rounding up and rounding down to the corresponding limits, Eq. 26 can be obtained. Finally, considering $\Delta^g \in [0,1]$, Eq. 26 become Eq. 27, which is consistent with Lemma 2.
>
> ## Re 3. Terminology[Experiment,W1]
> Thank you for the constructive suggestion. We will change "pixel" to "point" when coresponding to the DCS kernels in the final version.
>
> ## Re 4. Explanation of Figure 1.[Experiment,Q2]
> Thank you for your constructive comments. The circle marked as $\mathbb{K}$ in Figure. 1 represents the potential distribution of all DCS kernels. DCS automatically samples a random kernel from K at each forward propagation.
>
> ## Re 5. Connections to other relevant domains.[RelatedWorks,Q4]
> Thank you for your valuable comments. We have compared DCS against other stochastic adversarial defenses methods in Table 2. We divided the stochastic adversarial defenses into random weights and random structures. We are happy to add certified defenses and stochastic activation functions to the table and cite related work. Table 2 will then be expanded as the following table (only expanded part and ours).
> |Type|Method|CIFAR10||Imagenet||
> |:-:|:-:|:-:|:-:|:-:|:-:|
> |||PGD|AA|PGD|AA|
> |Ceritied Defense|Cert-RA[1]|68.6|-|-|-|
> |Random Structure|stochastic activation functions[2]|67.4|-|-|-|
> ||DCS (ours)|**75.84**|75.46|52.38|66.79|
>
> ## Re 6. Is Step 4 trained alternately or in a unified manner?[Q1]
> Thank you for your valuable questions. To avoid misunderstanding, we first claim that the randopm sampling processes in Figure 1 are not trainable.
>
> For the sampled kernels, we trained each kernel alternatively. Only one kernel is trained during each forward propagation, regardless of whether it is normal adversarial training or GSAT. Therefore we consider this to be alternating training.
>
> ## Re 7. Would combining DCS with adversarial training further improve robustness?[Q5]
> Yes. AT will further improve robustness. Corresponding results were compared in our experiments by Sec. 5.3.2 and Fig. 4(a). The results show that using AT or GSAT imporves the performance of DCS.
>
> ### Reference
>
> [1] Certified Defenses for Adversarial Patches, ICLR 2020.
>
> [2] Adversarial Defense Via Data Dependent Activation Function and Total Variation Minimization, ICLR 2019.

---

> > ### Comment · Reviewer_7KWx · 2025-04-03
> >
> > The authors have addressed my concerns. Thus, I increase my scores by one.

---

> > > ### Author Response · Authors · 2025-04-03
> > >
> > > We greatly appreciate your interest in the proof of theory and your time. It helped to make this paper more theoretically complete.

---

### Official Review · Reviewer_hpQK · 2025-03-13

**Overall Recommendation:** 3

**Summary:**

This paper proposes a random structural defense method called Deformable Convolution with Stochasticity (DCS) to improve adversarial robustness of convolutional neural networks. DCS replaces fixed convolutional kernels with randomly sampled deformable kernels to reduce adversarial transferability between inference paths in a data-independent way. The authors theoretically analyze the trade-off between robustness and clean accuracy in DCS and propose a Gradient-Selective Adversarial Training algorithm to further enhance robustness.

**Claims And Evidence:**

The main claims are mostly supported by adequate evidence in terms of theoretical derivations and experimental results. However, the claim of generalization and data independence equires more empirical support beyond CIFAR and ImageNet.

**Essential References Not Discussed:**

The related work section covers the most relevant prior work.

**Experimental Designs Or Analyses:**

The experimental setup for evaluating adversarial robustness against PGD and AutoAttack is mostly sound.

**Methods And Evaluation Criteria:**

The white-box robustness evaluations against PGD and AutoAttack are reasonable, but additional experiments against black-box and especially adaptive attacks would give a more complete picture.

**Other Comments Or Suggestions:**

The paper makes some good contributions in developing a randomized structural defense and providing theoretical insights, but has significant limitations in terms of generalization experiments, attack evaluations, analysis of computational efficiency and novelty.

**Other Strengths And Weaknesses:**

Weakness:
1. Rather than PGD and AA, the paper lacks sufficient evaluation against other important and updated classes of attacks. In particular, more results are needed against common transfer-based and query-based black-box attacks, adaptive attacks specifically designed for randomized defenses, and attacks that incorporate adversarial examples in the training set.

2. While the data-independent framework is a key advantage, the paper lacks sufficient empirical evaluation of DCS's generalization to different datasets beyond CIFAR-10/100 and ImageNet. More extensive experiments on a diverse range of datasets would be necessary.

3. The computational efficiency and training/inference cost of DCS is not sufficiently addressed. Replacing fixed convolutions with randomly sampled deformable convolutions could incur nontrivial computational overhead.

**Questions For Authors:**

No

**Relation To Broader Scientific Literature:**

The authors discuss several categories of related methods, including input/feature randomization, structure randomization, and stochastic networks. However, connections to other relevant domains like certifiable defenses could be drawn.

**Theoretical Claims:**

I checked the proofs in Appendix A.

---

> ### Author Rebuttal · Authors · 2025-03-31
>
> Your expert comments are constructive for our paper. We summarize and rebut your 4 major concerns in your comments.
>
> ## Re 0. Claim of generalization and data independence equires empirical support.[Claims,W2]
> Thank you for suggesting additional experiments to validate our claim. To verify the sensitivity of DCS hyperparameters to more distributions, we extended our experiments on STL-10[1] using ResNet18 as the baseline. The input size keeps the same as raw image as $96$. The results are shown in the following table:
> |Method|Clean|PGD|AA|
> |:-:|:-:|:-:|:-:|
> |baseline|61.95|38.65|35.49|
> |DCS|**62.40**|**42.25**|**47.95**|
>
> This results show the effectiveness of DCS on different distributions without changing the settings. We will include this experiment in the final version.
>
> ## Re 1. Additional experiments against attacks.[Evaluation,W1]
> Thank you for helping us to refine our experiments.
>
> **(1)** We conducted adversarial robustness tests on CIFAR-10 using ResNet18 for query-based, transfer-based and adaptive attacks that you mentioned. We expanded the results to Table 6 as
> |Model|Clean|SQUARE(query-based)|Pixel(query-based)|trnasfer-FGSM(transfer-based)[2]|transfer-FGSM-base(transfer-based)[2]|BPDA(adaptive)[3]|BPDA+EOT(adaptive)[3]|
> |:-:|:-:|:-:|:-:|:-:|:-:|:-:|:-:|
> |RN18|89.11|80.53|86.32|84.95|65.71|78.66|77.78|
> |WRN34|90.58|82.45|86.31|90.53|68.34|80.03|80.47|
>
> In the table, SQUARE and Pixel attacks are included in the initial paper. We added 4 columns to the table. The added attacks used the following settings:
>
>     transfer-FGSM:          base model=pretrained WRN50_2, epsilon=16/255
>     transfer-FGSM-base:     base model=pretrained baseline model, epsilon=8/255
>     BPDA:                   epsilon=8/255，max steps=20, learning rate=0.5
>     BPDA+EOT:               epsilon=8/255，max steps=20, learning rate=0.5, EOT steps=3.
>
>
> It can be seen that DCS appears to be robust against black-box attacks. We attribute this to the fact that the random masking of the convolutional kernel by DCS also shields a portion of the adversarial perturbations. For ataptive attacks like BPDA and EOT-BPDA, DCS also shows high robustness. We attribute this to randomness in conjunction with gradient masking.
>
> We will use the expanded Table 6 alone and analysis in the final version.
>
> **(2)** For attacks where adversarial samples are added to the training set, some predefined adversarial samples are added to the training set in both traditional adversarial training(AT) and GSAT. Related results of normal AT and GSAT can be found in Table 1 and Table 3.
>
> ## Re 2. Connections to other relevant domains like certifiable defenses could be drawn.[RelatedWorks]
> Thanks for this nice suggestion. We will add certifiable defense to Table 2 in the final version as:
> |Type|Method|CIFAR10||Imagenet||
> |:-:|:-:|:-:|:-:|:-:|:-:|
> |||PGD|AA|PGD|AA|
> |Ceritied Defense|Cert-RA[4]|68.6|-|-|-|
> |Random Structure|DCS (ours)|**75.84**|75.46|52.38|66.79|
>
> ## Re 3. Concern about computational overhead.[Q1]
> Thanks for this nice concern. We recorded the training and reasoning times for DCS+GGSAT vs. baseline+AT. We choose RseNet18 as the baseline. For training, we use CIFAR-10 training-set (50000 examples) following the hyperparameters:
>
>     epochs: 200
>     batch size: 128
>     optimizer: SGD
>     weight decay: 5e-4
>     initial learning rate: 0.1
>     scheduler: multiste (lr/10 at epoch 60 and 120)
> For reasoning, we use CIFAR-10 test-set (10000 examples) with batch size=1024. We recorded the entire time cost for training and reasoning in the table below.
> |Model|Training Overhead/min|Reasoning Overheal/sec|
> |:-:|:-:|:-:|
> |baseline|213.47|5.94|
> |DCS|379.96|6.67|
>
> Despite the difference in training time, their time consumption during inference is very similar. The time gap in training is due to the introduction of additional parameters in the DCS layer. However, during inference, the extra time consumption caused by the extra parameters is not as dramatic as training due to the absence of backpropagation.
>
> We are delighted to add this comparison in the appendix in the final version.
>
> ### Reference
>
> [1] An analysis of single-layer networks in unsupervised feature learning, Journal of Machine Learning Research - Proceedings Track 15, 215–223 (01 2011).
>
> [2] Explaining and harnessing adversarial examples, in International Conference on Learning Representations, 2015.
>
> [3] Obfuscated Gradients Give a False Sense of Security: Circumventing Defenses to Adversarial Examples, ICML 2018.
>
> [4] Certified Defenses for Adversarial Patches, ICLR 2020.

---

> > ### Comment · Reviewer_hpQK · 2025-04-03
> >
> > Thanks for this response and additional experiments. Most of my concerns have been addressed, so I have increased my overall rating.

---

> > > ### Author Response · Authors · 2025-04-03
> > >
> > > We greatly appreciate the addition of your experiments and your time. Your comments have helped us to present a more complete picture of DCS.

---

### Official Review · Reviewer_DMZf · 2025-03-14

**Overall Recommendation:** 4

**Summary:**

This paper introduces deformable convolution with stochasticity (DCS) to enhance the adversarial robustness of deep neural networks. Unlike traditional random defense methods that inject randomness into input data, this work incorporates randomness directly into the network architecture by replacing fixed convolutional offsets with random masks. Through a theoretical analysis of the trade-off between robust accuracy and natural accuracy, the authors identify kernel size as a key factor in balancing this trade-off. Additionally, the paper proposes a new adversarial training strategy that enhances performance by selectively masking pixels. Experimental results across multiple datasets demonstrate that DCS achieves superior adversarial robustness and clean accuracy compared to existing baselines.

### update after rebuttal
My concerns are well addressed during the rebuttal. I have updated my rating to 4: accept.

**Claims And Evidence:**

Yes, the claims are well-supported. The authors provide both empirical evidence and theoretical analysis to substantiate their findings.

**Essential References Not Discussed:**

No critical references appear to be missing.

**Experimental Designs Or Analyses:**

Yes, the experimental design and analyses are reasonable. However, there is a potential missing aspect regarding the experimental analysis of stride in Eq. 12. While the authors mention that S is set to a fixed value to ensure the same output feature map dimensions, downsampling layers exist in networks such as ResNet-18. These layers could be replaced by DCS to conduct a more thorough ablation study on the impact of stride.

**Methods And Evaluation Criteria:**

Yes, the proposed method is well-motivated by the theoretical analysis. The evaluation criteria are consistent with those widely adopted in adversarial robustness studies.

**Other Comments Or Suggestions:**

None.

**Other Strengths And Weaknesses:**

Strengths
1. This paper introduces an innovative strategy by embedding randomness into the network architecture rather than relying on data augmentation or noise injection. This design addresses key limitations of traditional random defense methods, such as data dependency and hyperparameter sensitivity.
2. The analysis of the trade-off between robust accuracy and natural accuracy provides valuable theoretical insights, potentially guiding the design of future random defense mechanisms.
3. The experiments convincingly demonstrate that DCS outperforms existing random defense methods in both adversarial robustness and natural accuracy. The consistency of results across multiple datasets and architectures further reinforces the effectiveness of the approach.

Weaknesses
1. It is evident from Table 3 that GSAT is more unstable than standard adversarial training (AT). However, the paper does not provide a clear explanation for this instability. Since Algorithm 2 should be responsible for stability, and the only difference between GSAT and standard AT lies in Algorithm 1, further clarification is needed. Additionally, the connection between this instability and Eq. 12 is not well-discussed.
2. The robust accuracy of Non-AT baselines is not reported in Figure 4(a). Including this information would provide a more comprehensive comparison.

**Questions For Authors:**

1. Can you conduct an ablation study by replacing downsampling layers in networks such as ResNet-18 with DCS to further analyze the effect of stride?
2. Can you provide a more detailed explanation regarding the instability of GSAT observed in Table 3 and its potential connection to Eq. 12?
3. Can you include the robust accuracy results of Non-AT baselines in Figure 4(a) for a more comprehensive comparison?

**Relation To Broader Scientific Literature:**

This paper builds upon research in deformable convolutions and random defense mechanisms. The application of deformable convolutions in adversarial robustness is a novel contribution, and the theoretical analysis of gradient similarity provides insights into the broader understanding of how randomness impacts model robustness.

**Theoretical Claims:**

Yes, I have verified the correctness of Lemma 1 and Lemma 2, and they appear to be sound.

---

> ### Author Rebuttal · Authors · 2025-03-31
>
> Thank you for your pertinent review and your interest. Your review is summarize and rebut by 3 major concerns.
>
> ## Re 0. Ablation study by replacing downsampling layers.[Experiment,Q1]
> This is an interesting problem. Your subtle experimental design help us to study the effect of stride in a limited network structure. We compare the results by replacing the DCS to the downsampled layer (**layer 6**) V.S. the closest normal convolution layer (layer 5, 7). The results are shown in the table below.
> |Layer|PGD|AA|
> |:-:|:-:|:-:|
> |5|68.12|70.72|
> |**6**|**72.03**|**75.74**|
> |7|69.36|72.11|
>
> We notice that the performance of DCS on downsampling layer is bettter than the normal convolution layers. According to the lemmas, larger sreide $S$ helps to reduce the assumed small numbers $\epsilon_c$ and $\epsilon_l$ in the lemmas. It results in a smaller gradient similarity and output distance, and finally increase the robustness and clean accuracy.
>
> In addition, larger $S$ also helps the bounds in both lemmas to be numerically looser in practice. This helps to find a suitable $n$ easier.
>
> ## Re 1. Explanation regarding the instability of GSAT in Table 3 and potential connection to the Equations.[W1,Q2]
> This is a constructive question. We will first explain the source of instability in GSAT and with Eq. 13, and then analyse the potential connection of Table. 3 with Eq. 12.
>
> **(1)** According to Eq. 13, GSAT modifies the sample space sampled by the DCS layer in forward propagation during training by selection. Below we explain the reason for the instability of GSAT shown in Table 3.
>
> We have demonstrated both theoretically and experimentally that paths in $\mathbb{X}^g$ and $\mathbb{X}^u$ are not easy to attack, while paths in $\mathbb{X}^s$ are very easy to be attacked by gradient-based algorithms (see Eqs. 7, 10 and Sec. 5.3.3 for details). This means that when the network is attacked, the gradients generated by the paths in $\mathbb{X}^g$ and $\mathbb{X}^u$ will be in different direction from the attacked gradients generated by the paths in $\mathbb{X}^s$.
>
> To minimize the influence of inaccurate attacked gradients in training, we select and remove paths in $\mathbb{X}^s$ from $\mathbb{X}$, and build the selected random space, as demonstrated in Eq. 13. This corresponds to step 11 in Algorithm 1. The selected random space is used for sampling paths in the forward and backward propagation for each training step sepreately.
>
> We understand that using GSAT the network are only limited optimized on the paths in $\mathbb{X}^s$ under each adversarial examples. So when the DCS samples paths in $\mathbb{X}^s$ in infernce, there are performance decrement compared to the other sampled paths. **The decrement will cause the instability in GSAT.** The larger probability the performance drop in DCS, the greater instability in GSAT.
>
> **(2)** For the relationship of results in Table 3 and Eq. 12, it mainly connected by $n$. From Table. 3, we can find that with larger $n$, GSAT will be more unstable. We give an explanation as: when Eq. 12 gives **larger bounds** for $n$, which refers to larger $n$ in practice, the percentage paths in unselected random space will increase. This increases the probability for DCS performance decrement in the networks using GSAT for training, which leads to greater instability. Thus, as shown in Table 3, $n$ is suggested to be small for a stable GSAT.
>
> The explanation of instability will be added in the appendix and the analysis of connection between Table 3 and Eq. 12 will be added in experiment section in the final version to helping audiances in understanding the nature of GSAT.
>
> ---
>
> In addition, we would like to argue that the paths in $\mathbb{X}^s$ are inherently difficult to optimize, given the difficulty of obtaining the correct gradients after a precision gradient-based attacked. Together with the results in Table 3 demonstrating that the network remains robust even with performance decrement. So we consider the performance decrement to be acceptable.
>
> ## Re 2. Unreported robust accuracy of Non-AT baselines.[W2,Q3]
>
> Thank you for your valuable suggestions.
>
> We supplemented the results of the NON-AT baseline in the table below. Cos in the table means gradient cosine similarity.
>
> |Model|Clean|PGD|Cos|
> |:-:|:-:|:-:|:-:|
> |baseline w/o AT|95.12|0.04|1.00|
>
> We will add these results as bars in Figure 4(a) in the final version.

---

> > ### Comment · Reviewer_DMZf · 2025-04-03
> >
> > Thanks for the detailed reply. Most of my concerns are addressed.

---

> > > ### Author Response · Authors · 2025-04-03
> > >
> > > We greatly appreciate your interest in the DCS and GSAT experiments and discussions and your time. Your comments have helped us refine our analysis of DCS and GSAT.

---

### Official Review · Reviewer_L7Yc · 2025-03-14

**Overall Recommendation:** 4

**Summary:**

This paper introduces Deformable Convolution with Stochasticity (DCS), a defense method that injects randomness into convolutional layers by replacing fixed offsets with random masks, thereby creating a data-independent random space for deformed kernels. This paper provides a theoretical analysis using gradient cosine similarity to derive strict, data-independent bounds on the receptive field. This paper further enhances this approach with Gradient-Selective Adversarial Training (GSAT), which selectively masks pixels with similar gradient origins to reduce adversarial transferability. Extensive experiments on CIFAR and ImageNet demonstrate that DCS with GSAT significantly improves both clean and robust accuracy compared with other random defense methods.

**Claims And Evidence:**

The claims made in the submission are well-supported by clear and convincing evidence.

**Essential References Not Discussed:**

None.

**Experimental Designs Or Analyses:**

The experimental design was evaluated primarily on CIFAR and ImageNet datasets using standard architectures like ResNet18 and WideResNet34, and included tests under multiple attack scenarios. I checked the design of these experiments and found that they are sound in terms of using established datasets and widely accepted attack methods for benchmarking adversarial robustness. However, as mentioned before, it is necessary to evaluate the proposed method using BPDA+EOT attack.

**Methods And Evaluation Criteria:**

In general, the evaluation criteria make sense for robust classification problems. For example, CIFAR-10, CIFAR-100 and ImageNet are widely acknowledged as benchmark datasets in this field. In addition, this paper uses a wide variety of attack methods to evaluate the robustness of baseline methods. However, since the proposed method introduces randomness, it is necessary to evaluate the proposed method using BPDA+EOT attack.

**Other Comments Or Suggestions:**

None.

**Other Strengths And Weaknesses:**

The proposed method is interesting and novel, which is significantly different from defense methods that inject randomness into data inputs. However, the proposed method is tailored for convolutional layers, and therefore it may require significant adaptations to be applied to other model architectures such as Transformers. Furthermore, defenses that incorporate randomness are vulnerable to adaptive attacks like BPDA [1] combined with EOT, which are designed to estimate the gradients despite the stochasticity. The proposed method should be evaluated against BPDA+EOT to demonstrate its robustness.

[1] Obfuscated Gradients Give a False Sense of Security: Circumventing Defenses to Adversarial Examples, ICML 2018.

**Questions For Authors:**

Regarding experimental design, how did you ensure that the randomness introduced by DCS and GSAT is properly controlled and that the robustness improvements are not a result of gradient masking? I am particularly wondering what is the robust accuracy of the proposed method under BPDA+EOT attack compared to the baseline methods. In addition, could you explain in more detail how the selective masking of pixels is performed during training and how this impacts the clean accuracy and robustness?

**Relation To Broader Scientific Literature:**

The paper builds on a rich body of literature in adversarial robustness by taking the idea of incorporating randomness to a structural level via deformable convolutions. Prior works (e.g., Xie et al., 2017; Li et al., 2019) have shown that randomness can hinder adversarial attacks, but they typically suffer from data dependency and require careful tuning of noise hyperparameters. This work extends those ideas by leveraging deformable convolutions (as introduced by Dai et al., 2017, and Zhu et al., 2018) to generate a randomized kernel space that is independent of input data. The proposed GSAT method also connects to and extends the body of research on adaptive adversarial training techniques by selectively mitigating gradient transferability issues, thereby refining existing defense strategies in a novel manner.

**Theoretical Claims:**

I reviewed the proofs provided for Lemma 1, which establishes the data-independent upper bound on the receptive field n, and the outline for Lemma 2 regarding the lower bound related to output inconsistency. The proof for Lemma 1 appears to be internally consistent and follows standard techniques in bounding gradient cosine similarity, though it relies on worst-case assumptions (e.g., setting Cg = 1) that simplify the analysis.  One potential issue is that both proofs assume certain distributions and independence properties of the gradients that might not hold exactly in practice. Overall, the proofs are mathematically plausible.

---

> ### Author Rebuttal · Authors · 2025-03-31
>
> Thank you for your expert comments and your interest in the content of our experiments. We summarize and rebut your 6 major concerns in your comments.
>
> ## Re 0. Evaluate under using BPDA+EOT attack.[Method,Experiment,W2,Q3]
> Thanks for this nice concern. We evaluated DCS under BPDA and BPDA+EOT attacks on CIFAR-10, following the hyperparameters:
>
>     epsilon=8/255，max steps=20, learning rate=0.5, EOT steps=3.
>
> In the final version, we will expand Table 6 to include:
> |Model|BPDA[1]|BPDA+EOT[1]|
> |:-:|:-:|:-:|
> |RN18|78.66|77.78|
> |WRN34|80.03|80.47|
>
> DCS is robust under BPDA+EOT attacks. We attribute this to randomness in conjunction with gradient masking.
>
> ## Re 1. Both proofs assume certain distributions and independence properties of the gradients that might not hold exactly in practice.[Theorem]
> Thanks for this nice concern. In practice, the data should be normalized. We assume the worst case of all **normalized** data to ensure that both lemmas stands in any data distribution after normalization. This does not mean that the worst case will necessarily occur. Therefore, in practice, the values of these two bounds is looser.
>
> We would like to emphasize that the worst case assumptions in proofs are specified on data distributions with infinite data points. This avoids the dependence of the bounds on specific data sets. This is the reason why the DCS setup can be data-independent.
>
> ## Re 2. Significant adaptations to apply DCS to other model architectures such as Transformers.[W1]
> Thanks for this nice concern. We believe this question reveals the future direction in randomized structure defense.
>
> **(1)** We confirm that DCS is tailor-made for convolutional operations. We believe that convolution is still an important tool in image processing. Transformer is out of the research scope of this work.
>
> **(2)** However, we briefly explored that DCS fits Vision Transformer(ViT). ViT uses a $16\times16$ convolution in patch embedding. Large kernel size makes lower bound of $n$ increases and hinders finding a suitable $n$. To avoid this, we notice that patch embedding can be split into multiple concatenated $3\times3$ convolutions[2]. Our baseline follows the settings in [2] and then replace the second $3\times3$ convolution with DCS. We obtained the following results
> |Method|PGD|
> |:-:|:-:|
> |ViT-t+Conv[2]|32.31|
> |ViT-t+DCS|**55.71**|
>
> DCS works well with ViT-tiny and we will add this experiment in appendix. We would like to argue that due to time constraints, the network is roughly trained on CIFAR-10 with two stages:
>
> **Stage 1**: Fix a pretrained ViT-tiny and train the FC layer and convolutions from scratch with:
>
>     -epochs: 200
>     -batch size: 128
>     -optimizer: SGD
>     -weight decay: 5e-4
>     -initial learning rate: 0.01
>     -scheduler: multiste (lr/10 at epoch 50 and 100)
>
> **Stage 2**: Adversarial finetune the entire network using GSAT with:
>
>     Stage 2:
>     -epochs: 90
>     -batch size: 128
>     -optimizer: SGD
>     -weight decay: 5e-4
>     -initial learning rate: 0.01
>     -scheduler: cosine
>
> ## Re 3. How to ensure that the randomness introduced by DCS and GSAT is properly controlled?[Q1]
> Thanks for this nice question.
>
> **(1)** For DCS, we control randomness by $n$ through Lemma1,2. Other setting in DCS are decided by the replaced convolution layer, to keep the data dimensions constant. The stride keeps the same. The kernel size is added by $2$, while padding is adapted accordingly.
>
> **(2)** For GSAT, the size of random space is smaller but **fixed** after removing the masks, since the number of removed masks is unchanged in each step.
>
> ## Re 4. How to ensure robustness improvements are not a result of gradient masking?[Q2]
> Thank you for your constructive comments on the additional experiment. To validate this, we manually cancel randomness. Instead, we selected two fixed deformable convolutional kernel $X^i$ and $X^j, n=4$ with no repeated points. $X^i$ is attacked and $X^j$ is used for reasoning. The results are
> |Method|PGD|
> |:-:|:-:|
> |DCN|52.23|
> |Fixed|54.71|
> |DCS|**62.93**|
>
> The robustness is mainly from randomness. We will add this observation into the appendix.
>
> ## Re 5. How is selective masking performed and how this impacts the clean Acc and robustness?[Q4]
> Thanks for this nice concern.
>
> **(1)** According to Algorithm 1, GSAT record the included points in the DCS kernel when generating the adversarial training examples. Then, all masks that unmask the recorded points are banned, until the end of current forward propagation. With the other masks, DCS will be trained with kernels sampled in $\mathbb{X}^g$ and $\mathbb{X}^u$.
>
> **(2)** From Eq(7,10), $\mathbb{X}^g$ and $\mathbb{X}^u$ helps to enlarge the gradient cosine similarity and minimize the output distances, which leads to higher robustness and clean Acc.
>
> ### Reference
>
> [1] Obfuscated Gradients Give a False Sense of Security: Circumventing Defenses to Adversarial Examples, ICML 2018.
>
> [2] Early convolu-tions help transformers see better, NIPS 2021.

---

> > ### Comment · Reviewer_L7Yc · 2025-04-03
> >
> > I would like to thank the authors for their comprehensive rebuttal. My major concerns have been well-addressed: 1), The authors provide additional results to show that their method performs well on BPDA+EOT (which is used to check gradient obfuscations). 2). The authors show that DCS fits Transformer, which demonstrates the generalizability of the proposed method. Therefore, I am willing to increase my score to 4.

---

> > > ### Author Response · Authors · 2025-04-03
> > >
> > > We are very grateful for your constructive comments and your time. Your suggested additional experiments on BPDA+EOT and transformer adaptation. This helps a lot in validation and generalizability of DCS. Your comment is instructive for the future study of stochastic structural adversarial defense.

---

### Decision · Program_Chairs · 2025-05-01

**Decision:**

Accept (poster)

**Comment:**

This paper creatively addresses adversarial robustness issues by designing a new module for neural networks. This contribution is new and interesting to the field of adversarial machine learning, which all reviewers agree. Although some concerns are raised in the initial comments, all of them are addressed by the authors during the rebuttal.

One limitation of this paper is that this paper considers CNN only (as one reviewer is concerned). However, this limitation cannot weigh too much based on its novel contributions to the field of adversarial machine learning.